# Population size changes and selection drive patterns of parallel evolution in a host–virus system

Jens Frickel[1,6], Philine G.D. Feulner[2,3], Emre Karakoc[4] & Lutz Becks [1,5]

Predicting the repeatability of evolution remains elusive. Theory and empirical studies suggest that strong selection and large population sizes increase the probability for parallel evolution at the phenotypic and genotypic levels. However, selection and population sizes are not constant, but rather change continuously and directly affect each other even on short time scales. Here, we examine the degree of parallel evolution shaped through eco-evolutionary dynamics in an algal host population coevolving with a virus. We find high degrees of parallelism at the level of population size changes (ecology) and at the phenotypic level between replicated populations. At the genomic level, we find evidence for parallelism, as the same large genomic region was duplicated in all replicated populations, but also substantial novel sequence divergence between replicates. These patterns of genome evolution can be explained by considering population size changes as an important driver of rapid evolution.

[1] Community Dynamics Group, Department Evolutionary Ecology, Max Planck Institute for Evolutionary Biology, 24306 Plön, Germany. [2] Department of Fish Ecology and Evolution, Center for Ecology, Evolution and Biogeochemistry, EAWAG, Swiss Federal Institute of Aquatic Science and Technology, 6047 Kastanienbaum, Switzerland. [3] Division of Aquatic Ecology and Evolution, Institute of Ecology and Evolution, University of Bern, 3012 Bern, Switzerland. [4] Wellcome Trust Sanger Institute, Genome Campus, Hinxton, Cambridge CB10 1SA, UK. [5] Kiel Evolution Center, Biologiezentrum, Am Botanischen Garten, 24118 Kiel, Germany. [6] Present address: KU Leuven Lab for Genetics & Genomics, Lab for Systems Biology, VIB Center for Microbiology, Leuven 3001 Heverlee, Belgium. Correspondence and requests for materials should be addressed to L.B. (email: lbecks@evolbio.mpg.de)

Parallel evolution is often observed when different populations or different species independently evolve similar phenotypes when adapting to the same environments[1]. Parallel evolution indicates a certain level of repeatability of evolutionary trajectories, which might seem counterintuitive in regard to the randomness at which mutations occur. Several experimental evolution studies have tested for the repeatability of evolution. Whereas some studies report that multiple replicate populations acquired the same adaptive solutions[2–6], others find divergent evolution where the replicate populations adapt differently to the same environment[7–10]. Predicting the repeatability of evolution and under which circumstances evolution results in diverging or parallel evolving populations remains uncertain[11, 12].

Generally, parallel evolution is more frequently found at the phenotype level, whereas evidence for parallelism is less common when considering the underlying genetic basis of the parallel phenotypes; parallel evolution is less frequently observed at the level of functional groups of genes, is reduced even further at the level of single genes, and almost never found at the base-pair level considering individual point mutations[13–15]. Thus, the degree of parallel evolution differs when looking at different levels of biological organization[16, 17].

Several population genetic parameters can affect evolutionary trajectories and thus patterns of parallelism and divergence between populations. Whereas strong selection might increase the probability of observing patterns of parallelism, genetic drift might lead to the opposite. For example, bottlenecks with a sudden and significant reduction in population size can affect divergence of populations due to genetic drift increasing the probability for fixation of mildly deleterious and effectively neutral mutations[18–20], as well as genetic hitchhiking of non-adaptive mutations, where mutations rise to high frequencies in the genetic background of beneficial variants that are selected[19, 21]. Differences in population sizes between populations exposed to similar conditions can also affect the degree of parallelism. For example, purging of maladapted genotypes is less efficient when intraspecific competition is reduced in small populations[22]. Beneficial mutations also occur more consistently in large populations[11, 19], where their fixation is more deterministic[23]. These and other population genetic processes can thus have significant effects on the degree and pattern of parallelism, potentially leaving signatures across the whole genome[18–20], at individual loci as well as at the phenotypic level.

Species interactions and especially antagonistic coevolution between consumer and resource populations are of particular interest for testing parallelism or divergence between populations and the repeatability of evolution because they typically exert strong selective pressures on each other, for e.g. ref.[24–26]. Antagonistic coevolution is further characterized by rapid and frequent changes of the selection regime, which can accelerate molecular evolution and potentially result in divergence of replicated populations over time[25–27]. Species interactions often lead to rapid and significant changes in population sizes and thus affect the likelihood of a mutation to occur and the probability of getting fixed. Changes in population sizes might either result from sheer ecological effects or from eco-evolutionary feedback dynamics[28–31], where ecology (e.g., population size) and evolution (e.g., selective sweeps) interact at the same time scale[31, 32].

Because ecological as well as evolutionary dynamics can affect divergence or parallelism on all levels of biological organization, a detailed understanding of the populations' evolutionary and ecological history is needed in order to understand the processes that drive parallel and divergent evolution. In this study, we investigate the degrees of parallelism and divergence in populations of an asexual reproducing eukaryotic algal host that coevolved with a dsDNA virus. We previously showed that replicated host and virus populations coevolve rapidly through arms race dynamics that directly affected host densities over time[31]. Bottlenecks in host densities occurred when none of the host individuals were resistant to the present virus, whereas the host population size increased after new resistant host types evolved (evolutionary rescue). In the present study, we compare the degrees of parallelism at different levels of organization between replicated host populations coevolving with the virus (hereafter: coevolved populations, $n = 3$) or evolving without the virus (hereafter: evolved populations, $n = 3$). We find parallel changes in population size and parallel evolution of host resistance in the replicate coevolving populations. On the genomic level, we find parallel evolution of a duplicated genomic region in these populations, but increased divergence in genome sequence between the replicate populations. We discuss that eco-evolutionary dynamics and the resulting demographic changes (changes in population sizes over time) modulate the pattern of evolutionary change on the phenotype and genotype level.

## Results

**Population size changes**. We ran continuous cultures (chemostats) for 90 days (~100 host generations) with the host *Chloralla variablis* alone (evolved populations, $n = 3$) and together with the virus strain Chlorovirus PBCV-1 (coevolved populations, $n = 3$). All chemostats were started from the same isogenic ancestor for the host (hereafter: ancestor population) and the virus. We observed significant cycles of algal and virus population sizes over time in all coevolved populations (Fig. 1a discussed in Frickel et al. 2016[31]). Specifically, we observed two bottlenecks in the algal population densities with rapid reductions to low densities, followed by rapid expansions before the populations stabilized and steadily increased in densities after day ~45 (Fig. 1a). We used wavelet coherence analyses[33, 34] to test for synchronization of the host population dynamics, i.e., significant parallelism of bottlenecks and population expansions. We looked at the time period from day 0 till day 45, as algal populations stabilized after day ~45 and thus no significant phase shifts could be identified hereafter. We indeed found significant correlations (alpha < 0.05) between the algal host population sizes over time with a phase shift of ~ zero (0.04 ±0.03 sd), meaning host densities changed in a very similar way over time. In contrast, the algal densities of the evolved populations were stable at their carrying capacity throughout the entire experiment (Fig. 1b).

**Coevolution**. Using time-shift experiments, we previously showed that host and virus were coevolving through arms race dynamics in the chemostat systems[31]. We found multiple cycles of hosts evolving resistance to virus, and virus evolving to infect previously resistant hosts again. Algal densities recovered after each bottleneck due to evolutionary rescue, i.e., the evolution and selective sweeping of new resistant host types, which correlated with increases in host population size (Fig. 1a). During the arms race, hosts became increasingly more resistant over time until a general resistant host type evolved around day 45. This general resistant host type is resistant to all virus types coming from the same population as the host, and is resistant not only to virus from its past but to virus types isolated from future time points[24, 31]. At the same time, the population dynamics of coevolved populations stabilized as a result of the general resistant type increasing in frequency. The host population at the end of the experiments consisted mostly of general resistant host clones (Fig. 2a; sympatric host–virus combinations), i.e., these hosts were resistant to all virus types (from their own replicate population) isolated from 11 time points from the start to the end of the experiment. Nevertheless, virus and less susceptible host types

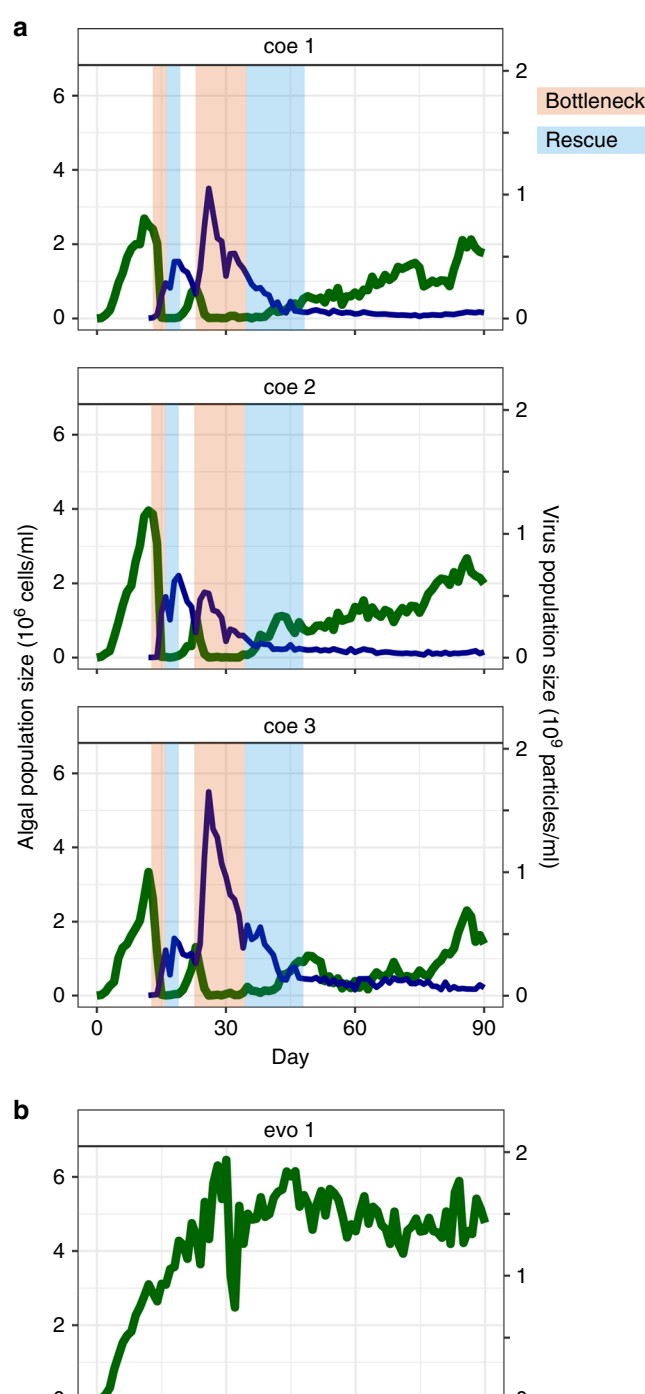

**Fig. 1** Parallel changes in host population size. Dynamics of three replicate algal-virus (coevolved, **a**) populations and one population with only algae (evolved, **b**), from continuous cultures over 90 days. **a** Host population dynamics showed two bottlenecks (orange-shaded) followed by rapid expansions (bue-shaded) in each replicate. **b** Algal population increased to high densities followed by stable densities around carrying capacity. Only one out of three replicates is shown

were maintained in the systems due to a trade-off between host resistance and growth rate[31].

**Parallel evolution of host phenotypes**. We tested whether the host–virus interactions in the coevolved populations resulted in

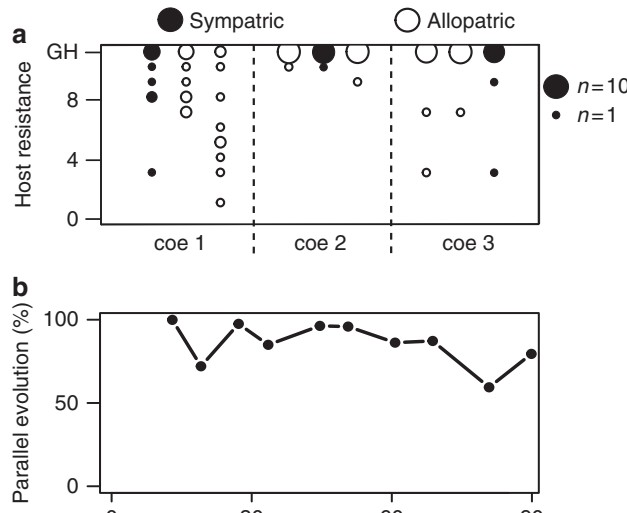

**Fig. 2** Phenotypic parallel evolution. **a** Host resistance (ranging from susceptible = 0 to general resistant host = GH) of 10 algal clones per replicate of the coevolved populations. Algal hosts were isolated from day 90 and for every replicate population and host resistance were tested to all virus types (isolated from 11 time-points, ranging from day 0 to day 90) from their own replicate (sympatric host–virus combination) and to all virus types form the other two replicates (allopatric host–virus combinations). The size of the dots correlates with how many hosts had that particular resistance range (total of 10 host clones per replicate, smallest dot size in figure equals to one host clone). Most algae were general resistant (sympatric host–virus combinations) and hosts that were general resistant were also general resistant when tested against all virus populations of the other replicates (allopatric host–virus combinations). **b** Degree of parallel evolution between the three replicates of the coevolved populations. Parallel evolution was calculated for 10 time-points ranging from day 12 to day 90. Hosts of the three replicate coevolved populations evolved in a highly parallel way over time (average = 87%; coe1–coe3 correspond to Fig. 1a)

parallel evolution of host resistance on the phenotypic level. Specifically, we asked whether the same resistant phenotypes evolved across all three replicates at the same time or whether the host–virus interactions resulted in divergent evolution with distinct resistant types across replicates. For this, we compared host resistance to sympatric viruses (isolated from their own local population) against their resistance to allopatric viruses (isolated from the other replicate populations). We tested sympatric and allopatric host resistance with 10 host clones for each of the 10 time-points spread over the course of the chemostat experiments and calculated the percentage of parallelism between the three replicate coevolved populations[35]. We found high levels of parallelism between the replicate populations with an average of 87% over all time points (Fig. 2b), which means that at each time point, most hosts that were resistant (susceptible) to virus from their own population were also resistant (susceptible) to virus isolated from the other populations at the same time point. Furthermore, knowing that most hosts from the last time point were general resistant hosts (Fig. 2a; sympatric host–virus combinations), we tested whether these hosts were also resistant to the virus types isolated from the other replicate populations. Indeed, general resistant hosts were also resistant to all virus types isolated from the different replicate chemostats (Fig. 2a; allopatric host–virus combinations).

**Evolution of host genotypes**. We used whole-genome sequencing to identify positions along the host genome that contained variants (single-nucleotide polymorphism: SNPs and small indels). We sequenced 10 individual isogenic host clones isolated from the last day of every replicate coevolved population (identical to the hosts used for resistance phenotyping Fig. 2a) and the evolved populations, as well as 10 individual isogenic clones from the population used to start the experiments (ancestor population). As ancestor clones were susceptible to the virus[31], we removed all positions that contained variants already present in the ancestor population for further analyses (total of 3871 positions over all populations). From these we compiled a data set consisting out of all derived variants, i.e., base-pair changes at the variable positions not found in the 10 ancestor clones (hereafter: 'derived variants'; 3927 over all populations as 56 positions had more than one alternative allele).

Overall, host clones from coevolved populations had significantly more derived variants compared with hosts from evolved populations (average number of derived variants per host clone from evolved: 114.4 ±48.8 sd; coevolved: 158.6 ±42.7 sd; comparison GLMER with and without evolution treatment as factor and replicate population as random effect: df = 1, $\chi^2$ = 4.41, p = 0.0357; family = negative binomial). We annotated and divided the impact of all derived variants into four classes: high (frameshift, splice donor variant), moderate (missense variant), low (synonymous variant), and non-coding (variant in intron or intergenic region). We found a significant difference in how the variants were distributed over the impact classes (Fig. 3a) between the evolved and coevolved populations (comparison GLM interaction between evolved vs. coevolved and impact class, df = 3, LRT = 77.468, $p < 2.2 \times 10^{-16}$, family = binomial) with proportionally more high-impact variants in the coevolved populations.

In order to identify patterns of parallelism across the three replicates we created a second data set containing only high frequency-derived variants. Hence, the data set of derived variants was further filtered for variants that rose in frequency to at least 50% in any evolved or coevolved population (hereafter: 'high frequency variants'). We choose a minimum of 50% because at least 50% of the clones in the coevolved populations were general resistant. Therefore, the high frequency variants can provide information about parallel evolution of resistance. Coverage was

relatively low in some regions (average ~8; Supplementary Fig. 1 for sequence quality statistics) and not all clones had sufficient coverage at every position to call a genotype (Supplementary Fig. 2). Therefore, a second requirement to identify variants for the high frequency data set was that at least six out of 10 genotypes could be called per population. This data set contained a total of 143 positions that had variants at relatively high frequencies in one or more of the evolved or coevolved populations (Supplementary Fig. 2).

The number of variants at high frequency in host clones from evolved compared to coevolved populations was not significantly different (average number of high frequency variants per host clone from evolved: 19.2 ±13.4 sd; coevolved: 26.1 ±11.1 sd; comparison GLMER with and without evolution treatment as factor and replicate population as random effect: df = 1, $\chi^2$ = 1.53, p = 0.216; family = negative binomial; Fig. 4) but variants were distributed significantly different over impact classes between the evolved and coevolved populations (comparison GLM interaction between evolved vs. coevolved and impact class, df = 3, LRT = 9.541, p = 0.023; family = binomial; Fig. 3b). However, this difference was largely owing to few high-impact variants occurring in two out of three evolved populations. Repeating the analysis without high-impact variants resulted in no significant difference between the coevolved and evolved populations (comparison GLM: interaction between evolved vs. coevolved and impact class, df = 2, LRT = 1.83, p= 0.40; with family = binomial). Furthermore, most high frequency variants were synonymous or were in introns and intergenic regions and had thus low impact.

**Parallel evolution of host genotypes**. The evolved and coevolved populations differed significantly in the proportion of derived variants that were unique per population. Most derived variants in the coevolved populations were unique and only occurred in one population. Differently, most of the derived variants in the evolved populations occurred in more than one population (comparison GLM with and without evolution treatment as factor: df = 1, LRT = 134.22, $p < 2.2 \times 10^{-16}$; family = binomial; 60 % unique in coevolved populations, 44% unique in evolved populations). This pattern was even more distinct when only considering high frequency variants. Approximately 50% of the high frequency variants identified in the coevolved populations

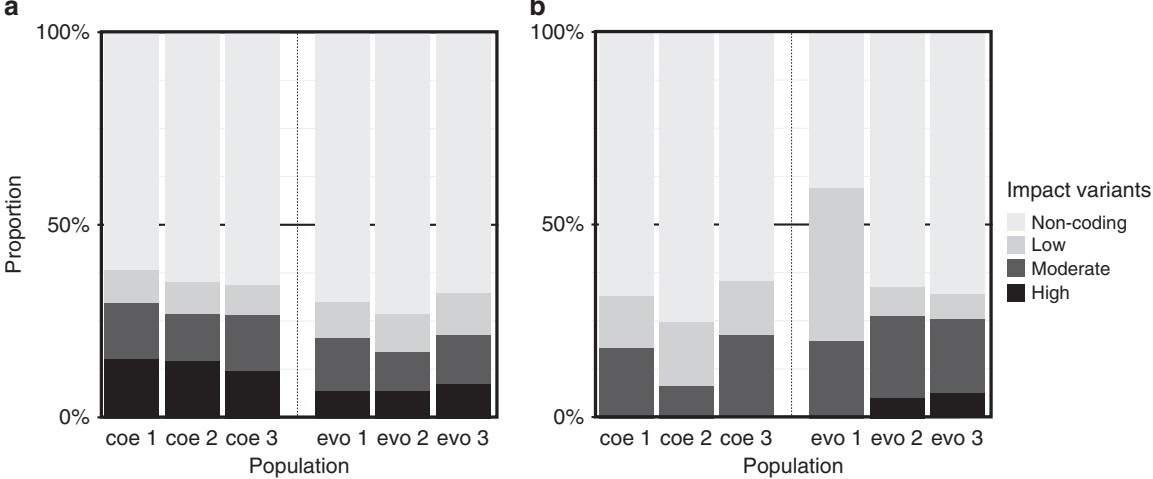

**Fig. 3** Relative abundance of variants with different impact effects. The proportion of variants per impact class for each population is shown. High = frameshift, splice donor variant, Moderate = missense variant, Low = synonymous variant, Non-coding = variant in intron or intergenic region. Abbreviations coe1–coe3 correspond to the three coevolved populations (Fig. 1a). Abbreviations evo1–evo3 correspond to the three evolved populations (Fig. 1b). **a** Distribution of all derived variants. **b** Distribution of high frequency variants (see Fig. 4)

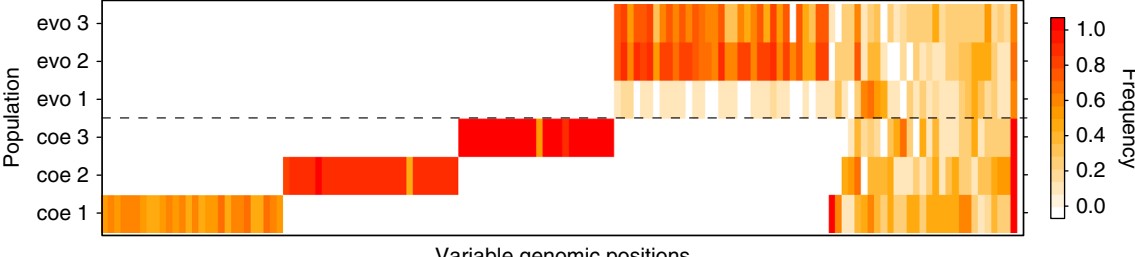

**Fig. 4** Frequency and distribution of high frequency variants across replicated populations for 143 variable genomic positions. Only high frequency variants (variants that evolved de novo, reached a frequency of 50% in at least one population, and the position had sufficient coverage in 6 out of 10 clones; see Results and Methods sections for more details) are plotted, with their corresponding frequency in the other populations. Each column represents a particular variable genomic position and positions were ordered to best visualize population specific and shared variants across populations. White color indicates no single clone out of 10 of this population had a variant at this position, yellow to red colors indicate increasing frequencies of the variant in that population. Total number of variants = 143. Abbreviations coe1–coe3 correspond to the three coevolved populations. Abbreviations evo1–evo3 correspond to the three evolved populations

were unique to one population, whereas almost all the high frequency variants identified in the evolved populations (99 %) were also present in at least one other evolved or coevolved population (comparison GLM with and without evolution treatment as factor: df = 1, LRT = 134.42 $p < 2.2 \times 10^{-16}$; family = binomial; 50 % unique in coevolved populations, 1% unique in evolved populations; Fig. 4).

We used the data set with high frequency variants to calculate the genetic distance between each of the three replicate populations based on Euclidean distance of frequencies to represent a general pattern of parallel evolution between populations. The evolved populations clustered significantly closer to each other than coevolved populations (average distance between evolved populations: 3.79, coevolved populations: 6.17; t-test, $t = -6.47$, df = 4, $p = 0.003$). Thus, divergence was greater between the three replicates of the coevolved populations compared to the evolved populations. Furthermore, the genetic distances calculated with all derived variants showed a similar significant pattern (average distance between evolved populations: 10.59, coevolved populations: 12.55; t-test, $t = -6.46$, df = 4, p = 0.003). Thus, greater divergence between coevolved populations was a general pattern of genome-wide sequence divergence.

Specifically looking at high frequency variants within genes, a total of 24 genes contained one or more high or moderate impact variants that were at high frequency in one or more populations (Fig. 5). Most genes that had high frequency variants in one of the coevolved populations did not have a variant in any other evolved or coevolved population. Conversely, all genes that contained at least one high frequency variant in one the evolved populations were also affected by variants in one or more of the other evolved populations and four genes had variants in both coevolved and evolved populations.

**Parallel evolution of a large genomic duplication**. We identified one distinct region in the genomes, for which all coevolved populations showed a significant increase in copy number compared with the ancestor and evolved populations (Fig. 6, Supplementary Fig. 3). Copy number increased in all coevolved populations in a shared region of ~77 kb. The exact extent of the duplicated region differed when comparing between the three replicates of coevolved populations (Fig. 6), but largely overlapped between the clones within every population. The duplicated region was detected in most clones in two coevolved populations (7 out of 10 clones, 9 out of 10 clones) but was less frequent in the third replicate (4 out of 10 clones; Fig. 6, Supplementary Fig. 3). Our data did not allow verifying whether this resulted from lower coverage in this population (Supplementary

Fig. 1), or whether this signal was genuine. Overall, the duplication was found in hosts that were general resistant. However, eight out of the 30 sequenced clones were not general resistant and six out of these hosts did have a copy number increase in this region. Interestingly, we found no evidence for copy number increase in the 10 sequenced ancestral clones in this region and only in two out of the 30 sequenced genomes from the evolved populations (Supplementary Fig. 3). This shared region contained 17 genes and our analysis did not show evidence for any other smaller variants in this segment. One of the coevolved populations had an additional duplicated region on a different scaffold that was at high frequency (Supplementary Fig. 3).

**Functional annotation of variants**. We annotated genes with high frequency variants and genes found within the shared duplicated region and compared coevolved and evolved populations regarding the gene functions identified (Supplementary Table 1). Most genes that had variants at high frequency in the coevolved populations had general metabolic functions, whereas most genes containing high frequency variants in the evolved populations were involved in cellular and signaling processes. Genes contained within the shared duplicated genomic region in the coevolved populations were roughly equally distributed within the three annotated general processes (for more information about potential functions of these genes and GO enrichment see Supplementary Tables 2–4).

**Discussion**

We investigated parallelism at different levels of biological organization between replicate populations of an algal host coevolving with a lytic virus and tested for the repeatability of their coevolution. The three replicates of the coevolved populations showed highly parallel changes in population sizes (Fig. 1a), with a similar timing of population bottlenecks and expansions. These changes in population size were largely driven by adaptation and counter-adaption of host and virus[31], indicating that evolution of resistance was a highly repeatable process. We further observed that evolution of host resistance was similar over all time points in the three replicate host populations (Fig. 2b). General resistant hosts evolved in all replicate coevolved populations and they were resistant to all virus types from their own coevolved population as well as to all virus types from the other coevolved populations (Fig. 2a). Thus, parallel evolution drove the coevolved populations to the same fitness peak with the same general resistant host phenotype.

Most of the coevolved hosts from the end of the experiments were generally resistant. Consequently, if the high levels of

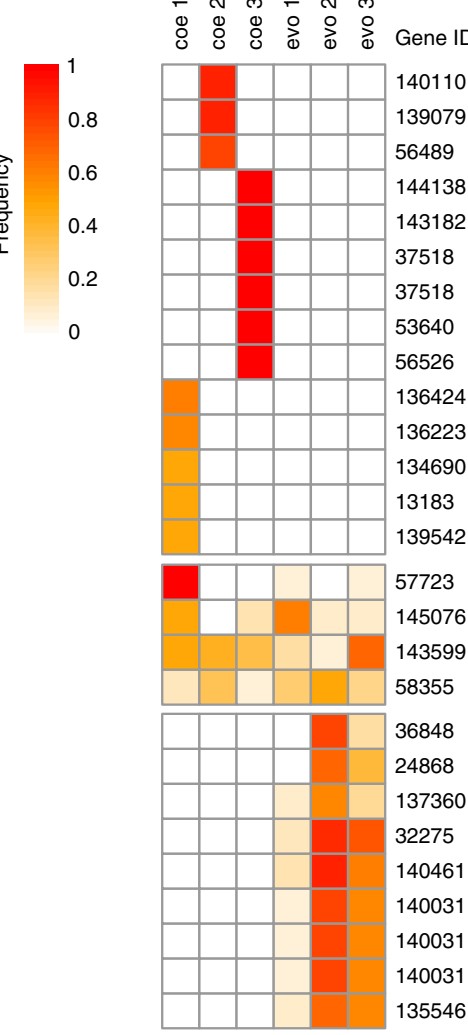

**Fig. 5** Frequency of non-synonymous high frequency variants distributed across 24 genes. Frequencies of non-synonymous and high frequency variants and their corresponding frequency in all populations. Each row is annotated with the gene ID that is affected by a particular variant. Four genes had variants both in evolved and coevolved populations. If genes had variants in one of the evolved populations, they were always found in at least another evolved population, and one gene (140031) had mutations at three positions at high frequency. In contrast, variants identified in the coevolved populations were specific to only one population, and one gene (37518) had mutations at two positions at high frequency. Abbreviations coe1–coe3 correspond to the three coevolved populations. Abbreviations evo1–evo3 correspond to the three evolved populations

parallelism observed on the phenotypic level of host resistance and population dynamics resulted from identical changes on the genotypic level, we would expect to find similar changes at the genomic level. Interestingly, we found a copy number increase of a large genomic region in all coevolved populations (Fig. 6 and Supplementary Fig. 3). This region had a different size in each replicate population, confirming that the duplication occurred in all three coevolved populations independently and was a highly repeatable evolutionary process. Duplications of genes have been shown previously in replicated bacteria lines adapting to antibiotics[36], in response to limiting resources or as compensation for deleterious mutations[37–39]. Generally, they are considered to be faster evolutionary responses than acquiring point mutations. Moreover, some regions of genomes are more receptive to duplications so that duplications can occur often at the same

genomic locations[36–38, 40–43]. Our observations support the idea that duplications can evolve readily within the same genomic region.

Some hosts from the coevolved populations were not generally resistant but did also carry the duplication and we found evidence for copy number increase in one of the evolved populations but only at very low frequency. It is thus unlikely that the duplication directly and solely invoked the evolution of general resistance, but the repeated evolution and rise in frequency of the duplication in the coevolved populations suggest a specific benefit during coevolution. Because the duplication remained at low frequency within the evolved populations, where no virus was added to the populations, it is likely that the duplication was not strongly adaptive under those conditions or evolved only at a later time points and hence did not reach high frequency.

We did not find evidence for parallel evolution between the replicate coevolved populations when looking at small variants (Fig. 4, Supplementary Fig. 2). Variants that occurred at high frequency in more than one population were also present in the evolved populations(Fig. 4), which were not exposed to the virus and did not evolve resistance (see Ref. [31] and Methods). Therefore, host resistance did not evolve by acquiring the same variants. Similarly, when looking at the level of genes, most genes that had non-synonymous high frequency variants were unique to one replicate of the coevolved populations (Fig. 5). Genes that acquired non-synonymous variants in more than one of the coevolved populations also had variants in the evolved populations. These results indicate that selection for host resistance was not targeting variants within the same sites or genes. Therefore, the evolution of host resistance was not driven by same underlying genomic changes when considering small variants and unlikely driven by duplications alone.

Sequencing hosts from coevolved and evolved populations allowed us to further infer how coevolution with the virus affected parallelism or divergence between the populations. We found that coevolved populations diverged more compared to evolved populations and had significantly more derived variants and proportionally more derived variants with a high-impact (Fig. 3a). This is in agreement with previous studies showing that coevolution accelerates molecular evolution[26]. However, when looking at high frequency variants, the number of variants was not significantly different and we found the opposite pattern for the proportions of impact classes (Fig. 3b) with a greater proportion of high-impact variants in the evolved populations. These observations together do not corroborate the notion that coevolution directly accelerated molecular evolution in our study system. Several recent studies show, however, that synonymous mutations can have direct effects on fitness and might therefore also contribute to evolutionary change over time[44–46]. However, evaluating potential fitness effects of mutations in more detail is outside the scope of our investigation. Yet, it is striking that evolved populations had many high frequency variants with the same base-pair changes (regardless of their impact class) that occurred in every replicate population in parallel, whereas they did not occur (not even at low frequency) in any of the coevolved populations (Fig. 4), suggesting that distinct parameters affected parallelism and divergence on the genomic level in the two treatments.

Indeed, besides the mere presence of the virus, the changes in host population sizes over time of the coevolved populations differed greatly from the evolved populations. We observed at least two bottlenecks in the densities of the coevolved populations followed by rapid population expansions, whereas densities of evolved populations were high and stable (Fig. 1). Bottlenecks can reduce the amount of standing genetic variation[47] and could have removed potentially adaptive variants from the populations. Our

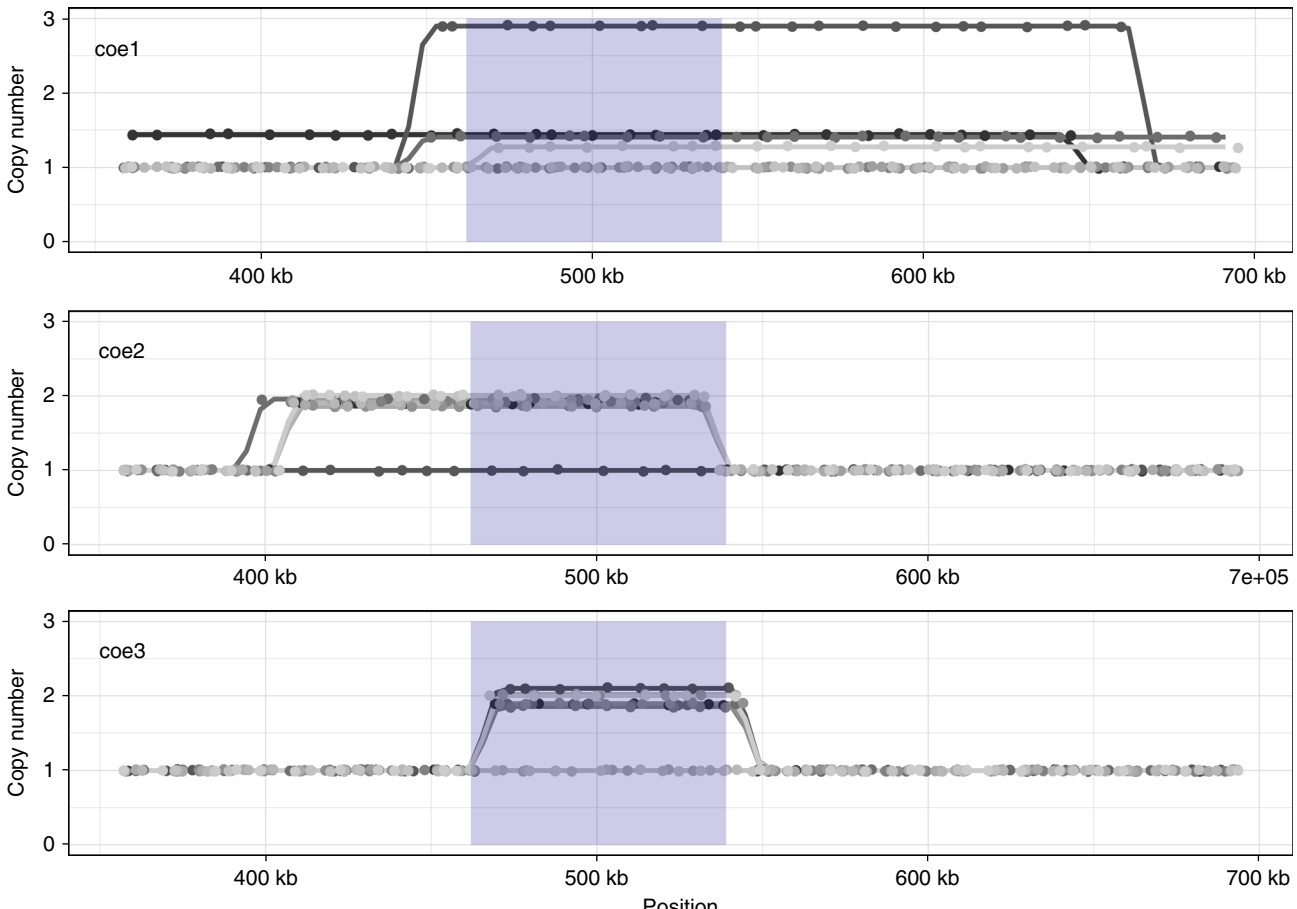

**Fig. 6** Large duplication partially overlapping in all three coevolved populations. For each of the three replicates copy number estimates for each of the 10 clones are plotted across a section of scaffold 24. Copy number was approximated for 1 kb window size. For clarity, points are estimates for every window of size 1 kb and are plotted with a random offset not to overlap the other replicates. Copy number increases for every coevolved population (coe1 = 4 out of 10, coe2 = 8 out of 10, coe3 = 9 out of 10 clones). Blue shading highlights the region where all three populations show a significant increase in copy number. Abbreviations coe1–coe3 correspond to the three coevolved populations

data show that, whereas many de novo variants rose to high frequencies in all the evolved populations in parallel, they were not found in any of the coevolved populations and might therefore have been removed from these populations by recurrent bottlenecks.

Furthermore, coevolution resulted in selective sweeps of new resistant host phenotypes and rapid expansions of the host populations[31]. Selective sweeps can result in random mutations hitchhiking in the genetic background with an adaptive mutation[19, 21]. The recurrent bottlenecks in combination with hitchhiking of variants in the coevolved populations likely led to the fixation of distinct sets of variants. We found that a large proportion of variants was different between the replicate coevolved populations (~50% unique high frequency variants). This pattern was very different in the evolved populations, where most of the high frequency variants occurred also in the other evolved populations (only ~1% unique high frequency variants), indicating that distinctive differences in ecological (population size changes) and evolutionary parameters (selective sweeps) determined the genomic evolution.

In addition, population densities at the end of the experiment were significantly lower in the coevolved populations. Therefore, purging of maladapted genotypes was likely not as effective and might have allowed for the accumulation of more variants at low frequencies, resulting in the significant larger number of derived variants in the coevolved populations compared the evolved

populations, as well as a higher proportion of high-impact variants in the coevolved populations at low frequency (Fig. 3). In summary, our data suggest that population bottlenecks, selective sweeps, and varying population densities left distinctive genomic signatures and affected the degree of parallel and divergent evolution between populations.

In agreement with previous studies, we show that the degree of parallel evolution differs when looking at different levels of organization. We found high levels of parallel evolution when looking at ecological (here: population size) changes and at the phenotypic level of resistance evolution. Population dynamics between the replicate coevolved populations were highly synchronized and coevolution led to the parallel evolution of general resistant hosts in different replicate populations. However, parallel evolution was not found at the genomic level of genes or variants (base-pair changes). Our data suggest that genomic divergence was likely driven by several population genetic parameters such as bottlenecks, sweeps, and population densities. Therefore, patterns of parallel and divergent evolution can be affected by strong interactions between ecology and evolution (eco-evolutionary dynamics), which are frequently observed during coevolution. Interestingly, the duplication of a large genomic region occurred in all coevolved populations in parallel. Additional investigations are needed to distinguish whether the evolution of the duplication was directly driven by coevolutionary

interactions and the duplication was adaptive in the context of host resistance, or whether another process was responsible.

## Methods

**Chemostat experiments.** Continuous flow-through experimental systems (chemostats, $n = 3$ per treatment) consisted of 500 ml glass bottles containing 400 ml of sterile Bold's basal medium (BBM) where nitrate was replaced by ammonium chloride. Sterile air and medium were supplied continuously at a rate of 10% per day. The cultures were maintained at 20 °C with continuous light and were mixed by stirring. One isogenic clone of *C. variabilis* was used to start all chemostat cultures. Purified and concentrated virus (Chlorovirus PBCV-1) was used to inoculate three replicates of the coevolved populations and three replicates of the evolved populations remained virus-free. For the latter we only followed population dynamics of one replicate in detail as previous experiments[31], in agreement with theory[48], showed that the algae alone grows at stable densities around the carrying capacity.

**Population dynamics.** Samples for assessing population densities were taken daily using standard sterile methods. Algal densities were enumerated by counting algal cells in life samples using a hemacytometer. Samples for assessing virus densities were filtered through a 0.45 μm cellulose syringe filter, the filtrate fixed with 1:100 gluteraldehyde and stored at −80 °C after freezing in liquid nitrogen. Daily virus densities were counted later by flow cytometry following Brussaard 2004[49] and Frickel et al. 2016[31]. We tested for a significant relationship between oscillations in algae populations of the replicates from the coevolution treatment. We did this by identifying the dominant and significant phase shifts between two time series using wavelet coherence analyses. This method measures the local correlation between two time series over a specific period and allows identifying significant and dominant phase shifts between two time series[33, 34]. We looked at the time period from day 0 till day 45, as algal populations stabilized after day ~45 and thus no significant phase shifts could be identified hereafter. The value of wavelet coherence is '0' when there is no relation between the two oscillations (no phase coupling) and '1' when there is a full correlation (perfect phase coupling) between the two oscillators. We extracted from these analyses the significant phase shifts (alpha < 0.05 and outside the cone of influence). Phase shifts show how one time-series lags behind the other; here algal population in one replicate behind algae population of the other replicate. A phase-shift around zero means that the two time-series cycle in phase (i.e., algal populations would increase and decrease at the same time), a phase shift of 1/2 means out of phase or anti-phase cycles (maximum is at the time of minimum of the other population, and vice versa). Significances of correlations were assessed by testing the null hypothesis that there is no correlation at a certain time of the time series by using a simulation algorithm representing white noise (default methods). We used the WaveletComp package[34] in R[50] and followed standard time-series analysis practices by de-trending (pracma package[51]) and smoothed time-series data (spline function in R[50]).

**Host resistance and quantification parallel evolution.** Host resistance range (sympatric host–virus combination: Fig. 2a) of hosts isolated at day 90 from the experiments was calculated as to how many virus populations from their own replicate a particular clone was resistant to. To do so, each host was tested against 11 virus populations separately sampled from different time-points from start to end of the experiment. Thus, a maximum resistance of 11 means these algal clones were general resistant (to all virus populations). During the experiments, virus samples were stored (at 4 °C after filtering through 0.45 μm cellulose filter) at regular time-intervals from the start of experiments to the end of the experiments (11 time points in total = 11 virus populations). Algae from the last day of the experiments were plated on agar plates and 10 random algal clones were picked from these agar plates and cultured in batch culture. Each algal clone was diluted to equal densities and challenged to the virus population (virus densities diluted to a multiplicity of infection of 0.01 particles/algal cell, four technical replicates per combination) from each time-point separately (10 algal clones×11 virus populations) in 96-well plates. Growth rates of algae exposed to the virus were calculated based on optical density measurements after 0 h and 72 h. To assess whether the algal clones were resistant or susceptible to a particular virus population, we compared the mean growth rate plus two standard deviations of four technical replicates to the mean growth rate minus two standard deviations of the control (host clone growth rates without virus). If the virus treatment value was equal to or greater than the control, these algae were considered resistant to this particular virus population. If the virus treatment value was smaller than the control, these algae were considered susceptible to this particular virus population. Allopatric host resistance range (Fig. 2a) was calculated similarly, but hosts were exposed to 11 virus populations (separately) isolated from the other coevolved populations.

Degrees of divergent and parallel evolution between the three replicate coevolved populations were calculated following Buckling and Rainey 2002[35]. We calculated degrees of parallel evolution for 10 time points (the same from which virus populations were isolated to calculate host resistance range above, but excluding the ancestor host population). To do so, algae from each of these time points were conserved on agar plates. From every time point, 10 random host clones were selected from the agar plates and grown in batch cultures. Each of these hosts was separately exposed to the virus population isolated from their own chemostat (and from the same time point from which that particular algae was isolated) and to the virus population isolated from the two other chemostats. Resistance and susceptibility of each algal clone was then assessed similarly as described above, and was used as a binary response variable and virus (from which replicate chemostat isolated), algal populations (from which replicate chemostat isolated) and their interaction as factors in a generalized linear model. The deviance explained by the main effects (deviance main effects/(deviance main effects + deviance interaction)) provided an estimate of the degree of parallel evolution, whereas the interaction provided an estimate of divergent evolution (deviance interaction/(deviance main effects + deviance interaction)). We previously showed (Frickel et al. 2016[31] and Frickel et al. 2017[24]) that algae isolated from the end point of chemostats where they evolve without the virus did not evolve any resistance against the ancestral virus. We thus did not further test whether or not the clones from the evolved populations evolved resistance or not.

**Genomic data and analysis.** We obtained whole-genome sequence reads by NGS (Illumina Nextseq 500 high throughput sequencing platform, high-output, read length = 150 bp, Nextera DNA Sample Preparation Kit; NextSeq 500 High Output Kit v2 (300 cycles)) of 10 individual isogenic clones coming from the last day (day 90) of every replicate of the coevolved (3 replicates × 10 clones = 30) and evolved (3 replicates × 10 clones = 30) populations and from the ancestor population (10 clones) that was used to start all the replicates. To isolate individual host clones, single colonies were picked from agar plates and grown briefly to sufficient densities in the same growth medium (modified BBM). Algal cells were concentrated by centrifugation, and potential bacterial cells were removed using a sucrose-density gradient. Algal DNA was extracted using cetrimonium bromide (CTAB)-DNA extraction method[52].

The sequences were trimmed (low quality leading and trailing bases below quality 20, headcrop 15 bases and maximum length 150 bases) and adaptor sequences removed using Trimmomatic[53] (URL: http://www.usadellab.org/cms/?page = trimmomatic). The reads were mapped to the reference genome[54] using the bwa-mem[55] (URL: http://bio-bwa.sourceforge.net/) tool with the default parameters and variants were identified after mark duplicates (picard[56], URL: https://broadinstitute.github.io/picard/) using standard GATK[57] pipeline via HaplotypeCaller (with the default options, ploidy set to one) and joint genotyping following the best practice for variant calling (https://www.broadinstitute.org/gatk/). In a first run, variants were only called in ancestor clones, and filtered manually with high quality SNPs (10 out of 10 clones annotated, allele frequency = 1, depth > 50, mapping quality > 50, fisher strand < 60, quality by depth > 2, MQ rank sum > −12.5, Read position rank sum > -8.0). These SNPs were used for recalibration of SNPs called in all coevolved, evolved and the ancestor samples and SNPs were filtered with a cutoff at tranche 90. We removed all variant positions found in the ancestor population from the data set containing all potential variant positions (from coevolved and evolved populations). INDELs were filtered manually using filtering parameters (quality by depth > 2, fisher strand > 200, Read position rank sum > −20, mapping quality > 50). Variants that were called in regions with clipped reads were removed. We tested for significant difference in number of variants (SNPs and INDELs) between coevolved and evolved populations using negative binomial models with replicate population as random effect (GLMER). A model with coevolved or evolved population as factor was compared to a model without the factor (null model). Similarly, we tested for a significant difference in the proportion of unique variants between coevolved and evolved populations by comparing a GLM (family = binomial) with coevolved and evolved population as a factor, to a model without the factor (null model). The genetic distance between coevolved and evolved replicate populations was calculated based on the frequency of variants using Euclidean distances in R. We tested for significant differences between the genetic distances using student t-test after testing and confirming equality of variances (based on potential adaptive variants: $F_{2,2} = 0.35$, $p = 0.52$; based on novel variants: $F_{2,2} = 0.039$, $p = 0.075$). The variants were then annotated using SnpEff[58] (URL: http://snpeff.sourceforge.net/, high, moderate,low and non-coding variants) and we tested whether there was a significant difference in how the variants were distributed in the different impact classes between coevolved and evolved populations. We tested whether the interaction between coevolved or evolved (as factor) with impact class (as factor) was significant by comparing a binomial GLM with the interaction and without the interaction.

In order to identify copy number, we used CNVnator[59] (URL: https://github.com/abyzovlab/CNVnator) with 1 kb sliding windows. We identified regions with significant copy number change by comparing copy numbers of each clone in every replicate population to the copy numbers of each clone in the ancestor population using pairwise t-testing corrected for multiple testing.

Functional annotation of genes containing high frequency–and high or moderate impact variants–was performed by searching for each gene (by protein ID = gene ID) the corresponding KOG ID, function and KOG Name by using the JGI Genome portal[60]. GO enrichment was performed using the Algal Functional Annotation Tool[61].

All statistics were performed using R version 3.3.2[50] and the lme4 package[62].

**Data availability.** The phenotypic and population density data that support the findings of this study are available in Dryad Digital Repository [doi:10.5061/

dryad.4gf1qb7][63]. The sequencing data have been deposited in the Sequence Read Archive [NCBI project accession no.: PRJNA450514].

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

## Acknowledgements

We are grateful to James Van Etten for providing us with the algal and virus cultures. This work was supported by an Emmy Noether Grant from the German Research Foundation (DFG) to L.B. (grant BE 4135/3-1). P.F. is supported by funding from the Swiss National Science Foundation (SNSF proposal number 310030E-160812).

## Author contributions

J.F., P.F., and L.B. conceived and designed the study, J.F. performed experiments, and all authors analyzed the results and wrote the paper.

## Additional information

**Competing interests:** The authors declare no competing interests.

