## [Peer Review File · Nature Communications]

Reviewers' comments:

Reviewer #1 (Remarks to the Author):

This study looks at parallel evolution across different levels of complexity (from phenotype to gene to base pair) and compares patterns of parallelism between "evolved" and "coevolved" populations. I thought this was a really interesting and novel study contained in a generally well-written manuscript. However I have a few concerns and suggestions that I outline below.

Main concerns:

- 1) My understanding is that the prediction that coevolution should accelerate molecular evolution is specific to adaptive mutations (and possibly the mutations that hitchhike with those adaptive mutations). In other words, variants would rise to high frequency in these populations and so be considered potentially adaptive by the authors' criteria. This prediction doesn't match with the results of the paper. Perhaps I am wrong in my understanding of the expectations - if this is the case, please set me straight. But if not, can the authors address this discrepancy?
- 2) It is not always clear in the paper when the authors are talking about parallel evolution at the gene level versus the base-pair level (e.g. on lines 187-198, and see my more specific comments below). Please be make sure to explicit about this.
- 3) As the authors discuss, population size is a potential driver of the different patterns of parallel evolution in the "evolved" versus "coevolved" populations. As such, they should provide those population sizes in the main text of the manuscript, and preferably in figure 1. Instead of a proportion of maximum size, plot actual population sizes - see more detailed comments below. I would like to know what the actual population sizes are and to gauge for myself how important I would expect the bottlenecks to be in terms of the potential for genetic drift.

Specific comments:

Line 100: "We ran continues cultures..." should read "We ran continuous cultures..."

Line 129: "... mostly out of general resistant host clones..." The word "out" should be removed here.

Lines 166-168: "In order to identify potentially adaptive variant positions, we filtered for variants that increased in frequency by at least 50% in any evolved or coevolved population." I don't understand what is meant here. I don't think the author's had data from different time points, so couldn't really measure increases in frequencies. Does this just mean any variant that was 50% or more of the population at the final time point?

Lines 187-189: "Differently, almost all the variants identified in the evolving populations (99%) were also present in one or more other populations at high or low frequency and where thus shared across the different evolved or coevolved populations." If I understand correctly, the authors are saying that 99% of the "potential adaptive" mutations *at the level of base-pair changes* arose multiple times independently in the evolving populations. I find this very surprising! Perhaps I am misinterpreting things?

Line 188: "... and where thus" should read "... and were thus..."

Lines 206-209: It would be useful to divide all the variants into impact classes and look to see if the distribution of classes varied between the "potential adaptive" and non-"potential

adaptive" variants differed. This would provide evidence as to whether or not the authors' approach for classifying "potential adaptive" variants was at all successful. Also, are the copy number variants (i.e the large duplications) included in this part of the analysis? If not, why not?

Line 218: "... that was on high frequency..." should read "... that was at high frequency..."

Lines 264-265: "We also found evidence for copy number increase in the evolved populations but only at very low frequency." Have the authors tested whether this evolved variant is also resistant to the viruses from the co-evolved communities? This might help to address the potential adaptive benefit of this duplication.

Lines 265-266: "Since the duplication remained at low frequency, it is likely that the duplication was not adaptive in case of the evolved populations." I'm not very convinced by this argument. The duplication is clearly adaptive in the co-evolutionary environment, and the authors state it is not involved in general resistance mechanisms. Perhaps it may be adaptive in a way that is unrelated to resistance, but it just hasn't had time to increase in frequency yet in this population.

Lines 273-276: "Moreover, certain regions of genomes are more receptive to duplications so that they can occur often at the same locations. Therefore, our observations support the idea that duplications can occur readily within the same genomic region." Do those regions that are more receptive to duplications have any identifying characteristics that the authors could look for in their own data?

Line 290: "...coevolved populations diverged greater ..." should read "...coevolved populations diverged more ..."

Lines 293-295: "Most of the potentially adaptive variants in the evolved and coevolved populations were synonymous substitutions or were in intergenic or intron regions (Fig. 4)." I am a little surprised by the sheer number of mutations that are apparently just hitchhiking on a presumably quite small number of beneficial mutations. Is it possible that these synonymous mutations are actually important for fitness?

Line 322: "... the pattern of less shared variants..." should read "... the pattern of fewer shared variants..."

Lines 396-397: I think this should read "If the virus treatment value was equal to or greater than the control, these algae were considered resistant to this particular virus population."

Fig. 1

From the caption: "Dynamics of algal (evolved, a) and algal-virus populations (coevolved, b)" The "a" and "b" are switched here.

Why only show a single rep for one treatment and 3 reps for the other treatment? I would suggest that the same number of reps should be shown for each.

Also, I think having the actual abundance on the y axes, would be much more informative. I'd like to see how similar the populations sizes between different replicates and treatments. The authors could have two y axis, one of the left and on the right, indicating the algae and the virus abundance respectively, allowing both abundances to fit nicely on the same graph.

Fig.4 – The purple and green boxes are unnecessary and distracting.

Fig 5 - I don't think this "combined frequency" measure is particularly easy to gain information from in this figure. It makes it difficult to work out what the frequencies are in each individual population. Why not just plot the population as points on a "frequency"

axis?

Reviewer #2 (Remarks to the Author):

Overview

--This manuscript explores aspects of parallel evolution using populations of *Chlorella* in replicate chemostats. In one treatment, *Chlorella* is grown alone. In another treatment, *Chlorella* is grown with an infectious virus population. Using population genetic and eco-evolutionary frameworks, the authors test for evidence of parallel evolution at demographic, phenotypic, and genotypic scales. Using wavelet analysis, the authors report that host densities respond similarly and synchronously in response to virus infection, which is used as evidence in support of demographic parallelism. The authors also characterize patterns of infectivity by challenging hosts against viruses from different chemostats across different time points. Findings from this approach support the view that there is parallel evolution at the phenotypic level. Evidence for parallel evolution at the genetic level was less clear, but the authors found an increase in copy number for a shared 77 kb region in the coevolved treatment. In contrast, the study did not find evidence for parallel evolution when looking at the frequency of variants across chemostats within a treatment.

Major comments

--While virus-induced bottlenecks could lead to the accumulation of mildly deleterious mutations, it would be worthwhile to look at how the divergent mutations in coevolved lines are distributed across the genome. If these mutations accumulate due to relaxed selection, then one would expect that they would be Poisson distributed across the genome. However, if these mutations are beneficial then they should be clustered within a few genes. To strengthen the argument that these divergent mutations are due to relaxed selection, it would be worthwhile to examine where these mutations are and how many of them are in the same region. This could be done using something similar to the G-score metric (see Tenaillon et al. 2016).

--The authors spend a large portion of the introduction talking about the hierarchy of parallel evolution and analyzed parallel evolution at the polymorphism, demographic, and infectivity levels (lines 43-38). However, they did not analyze parallel evolution at the gene level. More attention to this would strengthen the overall conclusions while serving as a bridge between the phenotype and genotype levels of parallel evolution. In the absence of such an analysis, there is a gap between the genotype and phenotypic components of the manuscript. For example, it is surprising that there is almost no discussion regarding the identity of "potentially adaptive" genes that confer virus resistance to the *Chlorella*. Figure 5 lists genes (e.g., 136424) that reaches a relatively high combined frequency in coevolved chemostats but is absent in evolved chemostats. Presumably these genes are annotated so that some discussion of potential function, which could be used to generate hypotheses to be tested in future studies. Moreover, a large portion of the manuscript (e.g., lines 226-243, 257-276) is devoted to the duplication of genomic regions. However, it is not clear what this region is or what it does. It would be useful for the reader to know what genes (if any) are in this genomic region.

--Throughout the paper, there are multiple references to novelty with regard to the integration of eco-evolutionary feedbacks into a parallel evolution framework. Such claims are overstated as this question has been addressed to varying degrees by other researchers (e.g., Bull 2005, Meyer et al. 2012; McElroy 2014; Perry et al. 2015).

--An average coverage of 8 is extremely low (line 168) and is a potentially serious issue. How might this affect the conclusions that are drawn? The GATK pipeline is tested and validated for 30X data and extremely low coverage could increase the number of false positives. There are some options that could be run in GATK to help with this. The authors did not state what options they used for the HaplotypeCaller. Setting --minDanglingBranchLength and --minPruning to 1.0 could help with the low coverage issues. Were there any extra steps taken or specifications set in HaplotypeCaller for calling SNPs in low-coverage isolates?

-- How are results and interpretation affected by 50% frequency when defining adaptive variants? What was the justification? Has a sensitivity analysis been conducted?

--The manuscript would benefit from a bit more polish in places. The Results section in particular was challenging to follow. The authors may want to consider how much weight should be given to each section based on testing of the predictions. It seems that some predictions could be addressed very succinctly. For example, even though wavelet test is rigorous, the population synchrony is almost assessable by eye. In contrast, the section on divergent evolution of host genotypes was long and less succinctly. And yet there were still many aspects of this data that did not seem adequately explored (see comments above). Also, good number of typos, which together with above, gave general impression that the manuscript could be a bit tighter in terms of data analysis and storyline.

-- "Generally resistant" is awkward and its use was confusing in many parts of the manuscript. Also, while I ultimately understood what was meant by "evolved" and "coevolved", perhaps this should be more clearly described with regard to treatments early in the manuscript.

Minor concerns:

Line 26: "demographic changes" initially unclear to me; simply means changes in abundance. Was wondering if there was going to be a more in-depth investigation into demographic processes

Line 29: "Can only be". wording always throws up red flags

Line 32: Should "where" be "when"?

Line 34: Mention of "convergent" evolution a red herring given aims?

Line 43: Selection acts to increase average fitness in a population, not sure that qualifies as parallel evolution.

Lines 51-54: The mutations are already random. A better way to phrase this would be "increasing the probability for fixation of mildly deleterious and effectively neutral mutations"

Lines 54-57: These two sentences could be integrated better. "Hitchhiking" is used before defining it.

Line 85: "We now compare here" is awkward phrasing. Also, wording in lines 89-91 very awkward.

Line 94: At this point reference to "variants" is unclear: hosts or viruses? Also line 116.

Line 107: Actually, not that remarkable.

Line 114: "at the 5% level". Awkward. Referring to $\alpha = 0.05$?

Line 116: "Differently" awkward adverb use.

Line 129: fix "mostly out"

Line 141: Perhaps some more clarification on metric of parallelism while keeping in mind that "local adaptation" sort of conjures up other ideas. Actually measuring lack of local adaptation.

Line 171: Why did the authors choose a cutoff of six out of ten? Is there any additional justification for this cutoff?

Line 178: fix "regardless their"

Lines 189-193: Same as previous comment. It sounds like measures should be taken to correct for potential false positives.

Line 200: fix "significant greater divergence"

Line 300: fix "were on"

Line 374: fix R45

Lines 460-464: It is not clear whether multiple testing correction was performed on the t-tests

Reviewers' comments in black, *our response in blue and italic.*

Reviewers' comments:

Reviewer #1 (Remarks to the Author):

This study looks at parallel evolution across different levels of complexity (from phenotype to gene to base pair) and compares patterns of parallelism between “evolved” and “coevolved” populations. I thought this was a really interesting and novel study contained in a generally well-written manuscript. However I have a few concerns and suggestions that I outline below.

We are pleased to hear that you evaluate our study interesting, novel, and well written.

Main concerns:

1) My understanding is that the prediction that coevolution should accelerate molecular evolution is specific to adaptive mutations (and possibly the mutations that hitchhike with those adaptive mutations). In other words, variants would rise to high frequency in these populations and so be considered potentially adaptive by the authors' criteria. This prediction doesn't match with the results of the paper. Perhaps I am wrong in my understanding of the expectations - if this is the case, please set me straight. But if not, can the authors address this discrepancy?

We fully agree about this predication and acknowledge that previously the presentation of our results has not been very clear in addressing the resulting discrepancies. For a start, we have changed the term 'potentially adaptive' to 'high frequency variants' in order not to confuse the reader. Further, we now clarify our prediction in line 93. We predict that under the assumption that molecular evolution is increased in the coevolving populations, we should find a higher number of variants that change amino acid sequence in hosts from coevolving populations compared to hosts from evolved populations. In the revised manuscript we now discuss this specifically by firstly considering all derived variants and afterwards only high frequency variants (see also paragraph in discussion at line 314). Furthermore, we do this for all variants and only for variants that change the amino-acid sequence (line 317-318). We now discuss that we find significantly more derived variants and proportionally more derived variants with a high-impact in the coevolved populations, which is in agreement with previous studies showing that coevolution accelerates molecular evolution (line 318-319). We then discuss that when only looking at high frequency variants (line 319-325), we observed a different pattern. We do not find significantly more high frequency variants, and we do find a greater proportion of high impact variants in the evolved populations. We follow this up by discussing how this observation can be explained (line 325-359).

2) It is not always clear in the paper when the authors are talking about parallel evolution at the gene level versus the base-pair level (e.g. on lines 187-198, and see my more specific comments below). Please be make sure to explicit about this.

We appreciate you pointing this out to us and have made this point more explicit in the revised manuscript. Specifically, we have explicitly stated whether we are talking about gene level or variant level (=base-pair level; referred to in line 163).

3) As the authors discuss, population size is a potential driver of the different patterns of parallel evolution in the “evolved” versus “coevolved” populations. As such, they should provide those population sizes in the main text of the manuscript, and preferably in figure 1. Instead of a proportion of maximum size, plot actual population sizes - see more detailed comments below. I would like to know what the actual population sizes are and to gauge for myself how important I would expect the bottlenecks to be in terms of the potential for genetic drift.

Thanks for pointing this out. We have changed Figure 1 accordingly and now provide information on the population sizes for the host and the virus.

Specific comments:

Line 100: “We ran continuous cultures...” should read “We ran continuous cultures...”

Changes accordingly.

Line 129: “... mostly out of general resistant host clones...” The word “out” should be removed here.

Changes accordingly.

Lines 166-168: “In order to identify potentially adaptive variant positions, we filtered for variants that increased in frequency by at least 50% in any evolved or coevolved population.” I don’t understand what is meant here. I don’t think the author’s had data from different time points, so couldn’t really measure increases in frequencies. Does this just mean any variant that was 50% or more of the population at the final time point?

We are sorry to hear that this did not come across clearly. As we have sequencing information from the algal clone used to start the experiment (=10 sequenced clones; see line 159), we are able to identify variants that newly arise after 90 days. So you are right we here refer to variants 50% or more at the end of the experiment, but absent at the start. Moreover, we have replaced the term ‘potentially adaptive variants’ with ‘high frequency variants’ in order to preclude further confusion.

Lines 187-189: “Differently, almost all the variants identified in the evolving populations (99 %) were also present in one or more other populations at high or low frequency and where thus shared across the different evolved or coevolved populations.” If I understand correctly, the authors are saying that 99% of the “potential adaptive” mutations *at the level of base-pair changes* arose multiple times independently in the evolving populations. I find this very surprising! Perhaps I am misinterpreting things?

Indeed, this result was also striking for us. We do agree that the term ‘potential adaptive’ mutations was misleading and changed this now to ‘high frequency variants’. We have now also discussed this point more elaborately in the manuscript by comparing this high level of parallelism to the much lower level in

the coevolved populations (line 327-331). We use this interesting aspect to further explain how differences in ecology (population sizes, bottlenecks; lines 332-340 & lines 351-359) and evolution (selective sweeps; lines 341-350) can drive such distinct differences between evolved and coevolved populations.

Line 188: "... and where thus" should read "... and were thus..."
Changes accordingly.

Lines 206-209: It would be useful to divide all the variants into impact classes and look to see if the distribution of classes varied between the "potential adaptive" and non-"potential adaptive" variants differed. This would provide evidence as to whether or not the authors' approach for classifying "potential adaptive" variants was at all successful. Also, are the copy number variants (i.e. the large duplications) included in this part of the analysis? If not, why not?
We agree that the use of 'potential adaptive' variants is misleading. We thus have changed the term and now use 'high frequency variants'. Although they might be potentially adaptive, several of these variants are likely at high frequency due to drift and population bottlenecks. We now also compare the distribution of classes between high frequency variants and all derived variants by including a new figure (Fig. 3) and discuss the observed patterns in more detail in the main text (line 314-331).

We would predict the copy number variant to have a high impact as it incooperates multiple genes (details see line 351). However, as we do not have specific phenotypic or molecular tests for this variant, we decided to not include the copy number variants with the other variants. Moreover, we found it interesting to compare single variants and copy number variation separately in terms of parallel or divergent evolution. We therefore discuss the duplication and single variants separately in the manuscript.

Line 218: "... that was on high frequency..." should read "... that was at high frequency..."
Changes accordingly.

Lines 264-265: "We also found evidence for copy number increase in the evolved populations but only at very low frequency." Have the authors tested whether this evolved variant is also resistant to the viruses from the co-evolved communities? This might help to address the potential adaptive benefit of this duplication.

We previously showed (Frickel et al. 2016¹ and Frickel et al. 2017²) and found in other independent experiments (Theodosiou & Becks unpublished data) that algae isolated from the end point of chemostats in which they evolve without the virus did not evolve any resistance against the ancestral virus. We thus did not further test whether or not the clones from the evolved populations evolved resistance or not. We state this now in the method section. We fully agree that this would be an interesting test, which unfortunately is not feasible anymore.

Lines 265-266: "Since the duplication remained at low frequency, it is likely that the duplication was not adaptive in case of the evolved populations." I'm not very

convinced by this argument. The duplication is clearly adaptive in the co-evolutionary environment, and the authors state it is not involved in general resistance mechanisms. Perhaps it may be adaptive in a way that is unrelated to resistance, but it just hasn't had time to increase in frequency yet in this population.

We are sorry about our confusing argumentation regarding this point. In our revised version, we rephrased (see lines 298-301) and this now reads "Because the duplication remained at low frequency within the evolved populations, where no virus was added to the populations, it is likely that the duplication was not adaptive or evolved only at a later time points and did not reach high frequency".

Lines 273-276: "Moreover, certain regions of genomes are more receptive to duplications so that they can occur often at the same locations. Therefore, our observations support the idea that duplications can occur readily within the same genomic region." Do those regions that are more receptive to duplications have any identifying characteristics that the authors could look for in their own data?

We agree that this would be a very interesting detail to look into. However, breakpoint resolutions are mostly still coarse. Hence we are not aware of any clear identifying characteristics we could look for and in which proximity of the variant one would need to look for it.

Line 290: "...coevolved populations diverged greater ..." should read "...coevolved populations diverged more ..."

Changes accordingly.

Lines 293-295: "Most of the potentially adaptive variants in the evolved and coevolved populations were synonymous substitutions or were in intergenic or intron regions (Fig. 4)." I am a little surprised by the sheer number of mutations that are apparently just hitchhiking on a presumably quite small number of beneficial mutations. Is it possible that these synonymous mutations are actually important for fitness?

Of course, synonymous mutations and especially variants in intergenic regions and introns might as well have phenotypic effects and could be fitness relevant. However, it is currently not possible to obtain knowledge for which and which proportion this might be the case. However, we agree with your evaluation and rephrased our manuscript (see line 324) to acknowledge this possibility: "Several recent studies show, however, that synonymous mutations can actually be important for fitness too and might therefore also substantially contribute to evolution (Bailey Nat.Comm. 2014, Agashe Mol.Biol.Evol. 2016, Lawrie PLOSgenet. 2013)."

Line 322: "... the pattern of less shared variants..." should read "... the pattern of fewer shared variants..."

Changes accordingly.

Lines 396-397: I think this should read "If the virus treatment value was equal to or greater than the control, these algae were considered resistant to this

particular virus population.”

Changes accordingly.

Fig. 1

From the caption: “Dynamics of algal (evolved, a) and algal-virus populations (coevolved, b)” The “a” and “b” are switched here.

Sorry for that mistake, we have corrected it.

Why only show a single rep for one treatment and 3 reps for the other treatment? I would suggest that the same number of reps should be shown for each.

We only followed population dynamics of one replicate in detail as previous experiments (Frickel et al. 2016 Ecology Letters) and in agreement with theory, showed that the algae alone grows at stable densities around the carrying capacity. We state this now in the methods section (lines 386)

Also, I think having the actual abundance on the y axes, would be much more informative. I’d like to see how similar the populations sizes between different replicates and treatments. The authors could have two y axis, one of the left and on the right, indicating the algae and the virus abundance respectively, allowing both abundances to fit nicely on the same graph.

We have changed Figure 1 accordingly, so in the revised version population sizes are shown as algal cells/ml (left y-axis) and virus particles/ml (right y-axis).

Fig.4 – The purple and green boxes are unnecessary and distracting.

We have removed the boxes.

Fig 5 - I don’t think this “combined frequency” measure is particularly easy to gain information from in this figure. It makes it difficult to work out what the frequencies are in each individual population. Why not just plot the population as points on a “frequency” axis?

We agree that this representation was not clear. We have changed Figure 5 and show the data now in a heat-map format.

Reviewer #2 (Remarks to the Author):

Overview

--This manuscript explores aspects of parallel evolution using populations of *Chlorella* in replicate chemostats. In one treatment, *Chlorella* is grown alone. In another treatment, *Chlorella* is grown with an infectious virus population. Using population genetic and eco-evolutionary frameworks, the authors test for evidence of parallel evolution at demographic, phenotypic, and genotypic scales. Using wavelet analysis, the authors report that host densities respond similarly and synchronously in response to virus infection, which is used as evidence in support of demographic parallelism. The authors also characterize patterns of infectivity by challenging hosts against viruses from different chemostats across different time points. Findings from this approach support the view that there is parallel evolution at the phenotypic level. Evidence for parallel evolution at the genetic level was less clear, but the authors found an increase in copy number for a shared 77 kb region in the coevolved treatment. In contrast, the study did not find evidence for parallel evolution when looking at the frequency of variants across chemostats within a treatment.

Major comments

--While virus-induced bottlenecks could lead to the accumulation of mildly deleterious mutations, it would be worthwhile to look at how the divergent mutations in coevolved lines are distributed across the genome. If these mutations accumulate due to relaxed selection, then one would expect that they would be Poisson distributed across the genome. However, if these mutations are beneficial then they should be clustered within a few genes. To strengthen the argument that these divergent mutations are due to relaxed selection, it would be worthwhile to examine where these mutations are and how many of them are in the same region. This could be done using something similar to the G-score metric (see Tenaillon et al. 2016).

We agree with the reviewer that this is an interesting aspect. We have now investigated and presented this aspect in more detail for the 'high frequency variants' (note that we changed our wording from 'potentially adaptive' to 'high frequency variants' in response to the reviewers comments). Figure 5 now shows that there are 2 genes that acquired more than 1 variant at relatively high frequency. One gene acquired two variants in one coevolved population, and one gene acquired more than one variant in all the evolved populations. However due to the relatively low number of non-synonymous mutations and the low number of multiple-hit genes (genes acquiring multiple throughout the experiment), we argue that a more in depth analyses of these genes and variants would not provide more insights regarding our main goal to investigate patterns of parallelism.

--The authors spend a large portion of the introduction talking about the hierarchy of parallel evolution and analyzed parallel evolution at the polymorphism, demographic, and infectivity levels (lines 43-38). However, they did not analyze parallel evolution at the gene level. More attention to this would strengthen the overall conclusions while serving as a bridge between the phenotype and genotype levels of parallel evolution. In the absence of such an analysis, there is a gap between the genotype and phenotypic components of the

manuscript. For example, it is surprising that there is almost no discussion regarding the identity of "potentially adaptive" genes that confer virus resistance to the *Chlorella*. Figure 5 lists genes (e.g., 136424) that reaches a relatively high combined frequency in coevolved chemostats but is absent in evolved chemostats. Presumably these genes are annotated so that some discussion of potential function, which could be used to generate hypotheses to be tested in future studies. Moreover, a large portion of the manuscript (e.g., lines 226-243, 257-276) is devoted to the duplication of genomic regions. However, it is not clear what this region is or what it does. It would be useful for the reader to know what genes (if any) are in this genomic region.

We agree that the reviewer raises an interesting point here. We have previously looked into this aspect mainly by presenting Figure 5, which shows genes and the number of variants within them with the corresponding frequency. In our revised version we have now further extend on this aspect by adding a table with the genes and their functions in the supplementary information (Supplementary Table 1), and summarized their function in general processes shown in Fig. 7. Moreover, we have used our set of genes to perform GO-enrichment analysis (summarized in Supplementary Tables 2,3). However, since no clear functional patterns were detected, we have chosen not to go into more detail about the functional annotation of the genes. Although this is certainly of interest for us, this manuscript rather focuses on the patterns of parallel evolution than on the explicit genomic mechanisms of resistance. Nevertheless, we provide the functional information now so that the reader can look at it and develop new hypothesis.

--Throughout the paper, there are multiple references to novelty with regard to the integration of eco-evolutionary feedbacks into a parallel evolution framework. Such claims are overstated as this question has been addressed to varying degrees by other researchers (e.g., Bull 2005, Meyer et al. 2012; McElroy 2014; Perry et al. 2015).

Thanks for pointing this out. We have removed our novelty statement.

--An average coverage of 8 is extremely low (line 168) and is a potentially serious issue. How might this affect the conclusions that are drawn? The GATK pipeline is tested and validated for 30X data and extremely low coverage could increase the number of false positives. There are some options that could be run in GATK to help with this. The authors did not state what options they used for the HaplotypeCaller. Setting --minDanglingBranchLength and --minPruning to 1.0 could help with the low coverage issues. Were there any extra steps taken or specifications set in HaplotypeCaller for calling SNPs in low-coverage isolates?

We agree that an average coverage of 8 is not very deep, however it is important to note that GATK recommendations refer to sequencing approaches for diploid organisms and specifically humans. However, here we sequence haploid algae clones, hence we are not concerned with identifying heterozygote genotypes, which indeed would warrant for a higher coverage. Meynert et al 2013³ estimated a local read depth of 13X is required to detect the alleles and genotype of a heterozygous single nucleotide variant (SNV) 95% of the time, but only 3X for a homozygous SNV. Hence, we think it is justified to run the GATK HaplotypeCaller using the default settings with the only exception of specifying ploidy as 1. This is now more clearly stated in the manuscript (see line 476). Please also note that we are further

making use of our biological replicates, by sequencing 10 clones for each treatment. Hence, especially our set of high frequency variants is expected to contain a minimal number of false positives.

-- How are results and interpretation affected by 50% frequency when defining adaptive variants? What was the justification? Has a sensitivity analysis been conducted?

We fully agree that the use of 'potential adaptive' variants is perhaps misleading for the reader. We thus changed the term and now use 'high frequency variants'. Although they are indeed potentially adaptive, several of these variants are likely at high frequency due to drift and population bottlenecks. We also agree that 50% frequency seems to be an arbitrary cut-off. However, this cutoff was chosen based on the knowledge that at least 50 % of the sequenced clones were generally resistant against all virus from their past. Hence, a subset of 'high frequency variants' matching the frequency of a resistant phenotype could therefore be more informative to investigate patterns of parallel evolution or general resistance. As suggested by the reviewer we have now also included a figure and detailed discussion of all derived variants, and compared our findings based on the high frequency variants with those based on all derived variants (e.g. Fig. 3). We think these comparisons provide a good insight for the reader regarding the sensitivity of the seemingly arbitrarily chosen cutoff. Moreover, we have also evaluated the effect using other cut-offs (sensitivity analysis, 30% - 50% - 70%) and included a figure summarizing the results in this response letter.

Figure 1. Different cut-off values for selection of 'high frequency variants'. Figure shows proportion and counts of variants within the 4 impact classes for selected variants based on different cut-off values.

--The manuscript would benefit from a bit more polish in places. The Results section in particular was challenging to follow. The authors may want to consider how much weight should be given to each section based on testing of the predictions. It seems that some predictions could be addressed very succinctly. For example, even though wavelet test is rigorous, the population synchrony is almost assessable by eye. In contrast, the section on divergent evolution of host genotypes was long and less succinctly. And yet there were still many aspects of this data that did not seem adequately explored (see comments above). Also, good number of typos, which together with above, gave general impression that the manuscript could be a bit tighter in terms of data analysis and storyline.

We appreciate the reviewer's feedback and have substantially edited the manuscript regarding readability of the result section, weighting of sections, correct typos and streamlined the storyline. We have further addressed the issues regarding data exploration the reviewer raised (see responses to the comments above). However, we do believe the manuscript is already complex, and therefore we did not go into detail for some aspects (e.g. GO-enrichment). However, we do provide the results of such analysis in supplementary information so that the reader can have a look if interested.

-- "Generally resistant" is awkward and its use was confusing in many parts of the manuscript. Also, while I ultimately understood what was meant by "evolved" and "coevolved", perhaps this should be more clearly described with regard to treatments early in the manuscript.

We used the term "general resistant" in previous publications (Frickel et al. 2016¹, Frickel et al. 2017²) and the term is often used in studies with antagonistic coevolving microbes (e.g. bacteria and phages). Therefore, we decided to keep the use of this terminology, but we have clarified our terminology further in the manuscript at lines 125 "This general resistant host type is resistant to all virus types coming from the same population as the host, and is resistant not only to virus from its past but also including those virus types isolated from future time points". We have now put more emphasizes on the explanation of evolved and coevolved and clarify these at the end of the introduction (lines 89) and at the start of the results (lines 103).

Minor concerns:

Line 26: "demographic changes" initially unclear to me; simply means changes in abundance. Was wondering if there was going to be a more in-depth investigation into demographic processes

We have changed it to "changes in population size" and changed our title accordingly. The title now reads: 'Population size changes and selection drive patterns of parallel evolution in a host-virus system'.

Line 29: "Can only be". wording always throws up red flags
We have tempered our statement.

Line 32: Should "where" be "when"?
Changed accordingly.

Line 34: Mention of "convergent" evolution a red herring given aims?
We removed our reference to convergent evolution.

Line 43: Selection acts to increase average fitness in a population, not sure that qualifies as parallel evolution.
We think this qualifies as parallel evolution as a phenotypically similar general resistant clones evolved in all three replicates.

Lines 51-54: The mutations are already random. A better way to phrase this would be "increasing the probability for fixation of mildly deleterious and effectively neutral mutations"
Thanks for the suggestion, we rephrased accordingly.

Lines 54-57: These two sentences could be integrated better. "Hitchhiking" is used before defining it.
We revised this section and now define hitchhiking first and attempted an improved integration of the two sentences. It now reads "...as well as genetic hitchhiking of non-adaptive mutations, where mutations rise to high frequencies in the genetic background of beneficial variants that are selected .." (lines 55-59).

Line 85: "We now compare here" is awkward phrasing. Also, wording in lines 89-91 very awkward.
We rephrased and it now reads "In the present study, we compare the degrees of parallelism ...".

Line 94: At this point reference to "variants" is unclear: hosts or viruses? Also line 116.
We say now: "...a higher number of variants that change amino acid sequences in the hosts from coevolved populations compared to hosts from populations evolving without the virus" (lines 93-96).

Line 107: Actually, not that remarkable.
We deleted remarkable.

Line 114: "at the 5% level". Awkward. Referring to $\alpha = 0.05$?
This now reads "We indeed found significant correlations ($\alpha < 0.05$) between the algal ..." (line 114).

Line 116: "Differently" awkward adverb use.
We removed differently, this now reads; "In contrast, the algal densities ..." (line 116).

Line 129: fix "mostly out"

We changed the phrasing, this now reads "... consisted mostly of general resistant ..." line 130.

Line 141: Perhaps some more clarification on metric of parallelism while keeping in mind that "local adaptation" sort of conjures up other ideas. Actually measuring lack of local adaptation.

We explain this now by saying: "We found high levels of parallelism between the replicate populations with an average of 87% over all time points (Fig. 2b), which means that at each time point, most hosts that were resistant (susceptible) to virus from their own population were also resistant (susceptible) to virus isolated from the other populations at the same time point." We also state: "we asked whether the same resistant phenotypes evolved across all three replicates at the same time or whether the host-virus interactions resulted in divergent evolution with distinct resistant types across replicates."

Line 171: Why did the authors choose a cutoff of six out of ten? Is there any additional justification for this cutoff?

We consider it necessary to make sure that at least six clones could be genotyped before calculating frequencies. We think this is an adequate decision for estimating frequencies. Estimating and reporting frequencies for positions with more missing data would skew our data set. Considering the effect of this choice, we might be missing parallel patterns, however we prefer to stay conservative and our approach makes sure that parallel patterns are not driven by missing data.

Line 178: fix "regardless their"

We deleted this sentence.

Lines 189-193: Same as previous comment. It sounds like measures should be taken to correct for potential false positives.

See response above.

Line 200: fix "significant greater divergence"

We removed this.

Line 300: fix "were on"

We removed this sentence.

Line 374: fix R45

We changed the reference accordingly.

Lines 460-464: It is not clear whether multiple testing correction was performed on the t-tests

We clarified this in the method section.

References

1. Frickel, J., Sieber, M. & Becks, L. Eco-evolutionary dynamics in a coevolving host-virus system. *Ecology Letters* **19**, 450–459 (2016).
2. Frickel, J., Theodosiou, L. & Becks, L. Rapid evolution of hosts begets species diversity at the cost of intraspecific diversity. *Proc. Natl. Acad. Sci. U. S. A.* **114**, 11193–11198 (2017).
3. Meynert, A. M., Bicknell, L. S., Hurles, M. E., Jackson, A. P. & Taylor, M. S. Quantifying single nucleotide variant detection sensitivity in exome sequencing. *BMC Bioinformatics* **14**, 195 (2013).

REVIEWERS' COMMENTS:

Reviewer #1 (Remarks to the Author):

I am happy with the changes that the authors have made in response to the previous reviews. I have only very minor wording/typo changes to suggest.

Lines 87-88: "... parallelism on different levels of organization..." Should read "... parallelism at different levels of organization..."

Line 253: "... duplicated region on a different scaffold that was on high frequency... " should read "... duplicated region on a different scaffold that was at high frequency... "

Line 279: "Most of the sequenced hosts from the end of the experiments were generally resistant." I suggest making this more clear by specifying "coevolved hosts".

Line 328: "(regardless their impact class)" should read "(regardless of their impact class)"

Reviewer #2 (Remarks to the Author):

By and large, the authors have addressed major concerns that were raised in the initial evaluation.

Some other issues that remain or now emerge include:

- The six out of 10 cutoff sounds a bit hand-wavy. However, it's only one of two criteria used for identifying variants.

- Figure 7 is fairly uninformative. It would be better to show the data in fig. 7 as a table and run some statistics to determine whether the proportion of mutations in a particular annotated region differ between evolved and co-evolved populations.

Minor issues:

line 29-31: grammar in second clause is awkward

line 32: revised ms reduced reference to demography, which is sort of vague, in favor of population size, which has more direct connection to evolutionary predictions; here is an example of where it is retained in a way that may be confusing; perhaps a legacy.

line 36: "Throughout evolutionary history" = weird; consider deleting

line 52: here and elsewhere "population genetics parameters"; consider changing to "population genetic parameters"

line 90-99: I appreciate concrete predictions, but it's not clear to me that reasonable

alternatives are considered or are appropriate, making me wondering whether the strengthens the paper or not.

Reviewers' comments in black, *our response in blue and italic.*

REVIEWERS' COMMENTS:

Reviewer #1 (Remarks to the Author):

I am happy with the changes that the authors have made in response to the previous reviews. I have only very minor wording/typo changes to suggest.

We are pleased to hear that you agree with the changes we made after the review and are thankful for your helpful feedback.

Lines 87-88: "parallelism on different levels of organization"; Should read "parallelism at different levels of organization";

We changed this sentence accordingly and also at line 272: "We investigated parallelism at different levels of biological organization between ...".

Line 253: "duplicated region on a different scaffold that was on high frequency"; should read "duplicated region on a different scaffold that was at high frequency";

We changed this sentence accordingly.

Line 279: "Most of the sequenced hosts from the end of the experiments were generally resistant"; I suggest making this more clear by specifying "coevolved hosts";

We changed this sentence; Most of the coevolved hosts from the end of the experiments were generally resistant.

Line 328: "(regardless their impact class)"; should read "(regardless of their impact class)";

We changed this sentence accordingly.

Reviewer #2 (Remarks to the Author):

By and large, the authors have addressed major concerns that were raised in the initial evaluation.

We are pleased to hear that you agree with our changes and are thankful for your helpful feedback.

Some other issues that remain or now emerge include:

The six out of 10 cutoff sounds a bit hand-wavy. However, it's only one of two criteria used for identifying variants.

We have addressed this concern in the previous evaluation:

"We consider it necessary to make sure that at least six clones could be genotyped before calculating frequencies. We think this is an adequate decision for estimating frequencies. Estimating and reporting frequencies for positions with more missing data would skew our data set. Considering the effect of this choice, we might be missing parallel patterns, however we prefer to stay conservative and our approach makes sure that parallel patterns are not driven by missing data."

In addition to this: The six out of 10 cutoff is a threshold chosen by us in order to ensure frequency data quality. This cutoff is calculated for each position and for each population. Therefore, if we could not

identify a particular variant in one population (because less than 6 genotypes could be called), it is likely that we could identify this position in the other 2 populations. Since we did not find strong evidence for parallel evolution even for 2 populations, we are confident that this cutoff does not affect our conclusions.

Figure 7 is fairly uninformative. It would be better to show the data in fig. 7 as a table and run some statistics to determine whether the proportion of mutations in a particular annotated region differ between evolved and co-evolved populations.

We agree and removed this figure from the main text. We included that data as an extra table in supplementary information (Supplementary information Table 1).

Minor issues:

line 29-31: grammar in second clause is awkward

We changed these lines to: "We find high degrees of parallelism at the level of population size changes (ecology) and at the phenotypic level between replicated populations. At the genomic level, we find evidence for parallelism, as the same large genomic region was duplicated in all replicated populations, but also substantial novel sequence divergence between replicates."

line 32: revised ms reduced reference to demography, which is sort of vague, in favor of population size, which has more direct connection to evolutionary predictions; here is an example of where it is retained in a way that may be confusing; perhaps a legacy.

We changed this line to: These patterns of genome evolution can be explained by considering population size changes as an important driver of rapid evolution.

line 36: "Throughout evolutionary history" = weird; consider deleting

We deleted this part of the sentence.

line 52: here and elsewhere "population genetics parameters"; consider changing to "population genetic parameters"

We changed this in line 54, line 66 and line 374.

line 90-99: I appreciate concrete predictions, but it's not clear to me that reasonable alternatives are considered or are appropriate, making me wondering whether the strengthens the paper or not.

We removed all predictions from the manuscript.